# Can a Single Tree Outperform an Entire Forest?

## Abstract

The prevailing mindset is that a single decision tree underperforms random forests in testing accuracy, despite its advantages in interpretability and lightweight structure. This study challenges such a mindset by significantly improving the testing accuracy of an oblique regression tree through our gradient-based entire tree optimization framework, making its performance comparable to random forests. Our approach reformulates tree training as a differentiable unconstrained optimization task, employing a scaled sigmoid approximation strategy. To ameliorate numerical instability, we propose an algorithmic scheme that solves a sequence of increasingly accurate approximations. Additionally, a subtree polish strategy is implemented to reduce approximation errors accumulated across the tree. Extensive experiments on 16 datasets demonstrate that our optimized tree outperforms random forests by an average of 2.03% improvements in testing accuracy.

## 1 Introduction

The single decision tree attracts significant attention in machine learning primarily due to its inherent interpretability. Its transparent "IF-THEN" decision rules make it highly useful for tasks that require clear decision-making logic behind predictions. However, its adoption is often limited by lower testing accuracy, particularly when compared to tree ensemble methods like random forests Breiman (2001). Random forests, built from multiple decision trees, are widely recognized for their superior testing performance over single tree models (Tan & Dowe, 2006), and are considered among the best models for accuracy (Fernández-Delgado et al., 2014; Grinsztajn et al., 2022). However, a random forest, which typically consists of hundreds of decision trees, diminishes—or even eliminates—the interpretability that a single decision tree provides. This trade-off between interpretability and accuracy has become widely accepted, fostering a common mindset that random forests outperforms single decision trees in testing accuracy, though at the expense of interpretability.

Such a mindset forces resorting to random forests when high test accuracy is essential, even in cases where a lightweight structure and interpretability are also demanded. For instance, in embedded systems with limited hardware resources and power budgets, a lightweight algorithm like decision tree is ideally preferable due to its fewer parameters and less energy consumption (Narayanan et al., 2007; Alcolea & Resano, 2021); yet, in practice, the subpar performance of a single tree often compels a shift toward tree ensembles, such as random forest (Elsts & McConville, 2021; Van Essen et al., 2012). This shift introduces two major issues: first, it significantly aggregates computational costs and memory consumption due to more parameters from multiple trees. Second, it sacrifices the interpretability, which is crucial in certain decision-support scenarios, such as piece-wise control law in explicit model predictive control (Bemporad et al., 2002) and threshold-based well control optimization in subsurface energy management (Kuk et al., 2022). These two concerns can be effectively addressed by a single decision tree if it could match the accuracy of random forests.

Aiming at a single tree with higher accuracy and fewer parameters, the oblique decision tree, a pivot extension of the classic orthogonal decision tree, holds great potential. Oblique decision trees use linear combination of features to create hyperplane splits. When the underlying data distribution follows hyperplane boundaries, oblique decision trees tend to simply tree structures, generating smaller trees with higher accuracy (Costa & Pedreira, 2023). Nevertheless, inducing oblique decision trees presents substantial computational challenges, owing to the innumerable linear combinations of features at each node (Zhu et al., 2020). Earlier works mainly focus on finding the optimal feature

combinations at an individual node using greedy top-down algorithms, such as CART-LC (Breiman et al., 1984) and OC1 (Murthy et al., 1994). Besides, alternative methods rely on greedy orthogonal decision trees CART to induce oblique trees by rotating the feature space, exemplified by HHCART (Wickramarachchi et al., 2015) and RandCART (Blaser & Fryzlewicz, 2016). Despite their advancements, such greedy methods that focus on optimal splits at current nodes, might lead to suboptimal solutions due to the weaker splits at subsequent child nodes. Considering the optimization of splits at all nodes, Bertsimas & Dunn (2017) presented optimal decision tree method to formulate tree training as a mixed-integer programming (MIP) problem. However, the practical application of MIP-based methods often face challenges in scalability and computational efficiency, especially in optimal oblique decision trees where the search space is expanded. Recent efforts in optimal oblique trees (Boutilier et al., 2023; Zhu et al., 2020) have been confined to classification tasks with a limited number of categorical prediction values. In contrast, addressing regression tasks with an infinite number of possible prediction values remains an extremely challenging task. In response to this limitation, the originators of MIP-based work further proposed an alternative local search method ORT-LS (Dunn, 2018) for tasks that are unsolvable by MIP. However, ORT-LS still suffers from high computational costs and suboptimal accuracy, as observed in our comparative studies.

In this work, we reformulate the training of an entire tree as an unconstrained optimization task, offering significant solvability advantages over MIP reformulations. This reformulation makes it easily solvable through exiting powerful frameworks of gradient-based optimization. Given the non-differentiability of indicator functions in hard splits, two intuitive solutions has been used in recent literatures: treating the gradient of those indicators as one via straight-through estimators (Karthikeyan et al., 2022; Marton et al., 2023) and approximating indicators with sigmoid functions (Wan et al., 2021; Yang et al., 2018; Frosst & Hinton, 2017). However, straight-through estimators may neglect crucial gradient information, resulting in suboptimal outcomes, as observed in both their work (Marton et al., 2023) and our experiments. In light of those works with sigmoid approximation, two major concerns arise. Firstly, previous efforts predominantly focused on constructing "soft" decision trees (İrsoy et al., 2012), characterized by soft splits and probabilistic predictions. Nonetheless, there do exist scenarios where a hard-split tree with deterministic predictions is not only appropriate but also imperative. Further, the probabilistic soft splits significantly deviates from the interpretable True-False, IF-THEN decision logic. Secondly, the simple use of sigmoid functions leaves a considerable gap from indicator functions, necessitating a delicate balance between approximation accuracy and numerical solvability by scaling the sigmoid function (Hehn & Hamprecht, 2017). However, identifying the optimal scale factor also remains a challenge. More importantly, less attention has been paid to the approximation error that can accumulate across the entire tree, particularly in deep trees with numerous nodes. These issues substantially degrade the testing accuracy of a gradient-based tree, far lagging behind the performance of random forests.

**Our contributions:** Firstly, we propose a strategy of iterative scaled sigmoid approximation to narrow the gap between the original indicator function and its differentiable approximation. This strategy uses the solution from an optimization task with a smaller scale factor to effectively warm-starts optimization with a larger scale factor. By starting with a smaller, smoother scale factor and gradually increasing it, this strategy enhance the approximation degree, while mitigating numerical instability typically associated with larger scale factors. Secondly, unlike soft trees with probabilistic predictions, we remain the hard-split decisions and deterministic predictions, only using soft approximation for gradient computations. Thirdly, to address severe approximation errors accumulated across each split in the entire tree, we propose a subtree polish strategy to further improve the training optimality. Finally, we provide an extensible **G**radient-based **E**ntire **T**ree optimization framework for inducing a tree with both constant predictions (termed as **GET**) and linear predictions (termed as GET-Linear), easily implemented in existing deep learning frameworks, as available in https://github.com/anonymweblinks/GET.

**Performance:** Experiments show that our method can produce a tree with testing accuracy comparable to, or even exceeding, that of random forests, challenging the prevailing mindset.

- Empirically and statistically, our oblique tree GET outperforms compared decision tree methods in test accuracy. Notably, it outperforms CART by 7.59% and the state-of-the-art ORT-LS by 3.76%.

- Our method GET statistically confirms its testing accuracy as comparable to random forests, and empirically underperforms random forests by a mere 0.24% gap.

- Our optimized oblique tree with linear predictions `GET-Linear` impressively outperforms random forests by an average of 2.03% in testing, demonstrating a statistically significant difference.

## 2 Foundations of Oblique Regression Tree

In this section, we explore oblique regression trees from an optimization perspective by formulating tree training as an optimization problem. For ease of understanding, we primarily follow the notation for optimal decision trees as used in the original work of Bertsimas & Dunn (2017).

Consider a dataset comprising $n$ samples denoted as $\{\boldsymbol{x}_i, y_i\}_{i=1}^n$ with input vectors $\boldsymbol{x}_i \in [0,1]^p$ and true output values $y_i \in [0,1]$. A binary tree of depth $D$ comprises $T = 2^{D+1} - 1$ nodes, where each node is indexed by $t \in \mathbb{T} = \{1, \cdots, T\}$ in a breadth-first order. The nodes can be categorized into two types: branch nodes, which execute branching tests and are denoted by indices $t \in \mathbb{T}_B = \{1, \cdots, \lfloor T/2 \rfloor\}$, and leaf nodes denoted by $t \in \mathbb{T}_L = \{\lfloor T/2 \rfloor + 1, \cdots, T\}$, responsible for providing regression predictions. Each branch node comprises a split weight $\boldsymbol{a}_t \in \mathbb{R}^p$ and a split threshold $b_t \in \mathbb{R}$ to conduct a branching test ($\boldsymbol{a}_t^T \boldsymbol{x}_i \leq b_t$) for the samples allocated to that particular branch node. If a sample $\boldsymbol{x}_i$ passes the branching test ($\boldsymbol{a}_t^T \boldsymbol{x}_i \leq b_t$), it is directed to the left child node at index $2t$; otherwise, to the right child node at index $2t + 1$. Each leaf node comprise the parameters of $\boldsymbol{k}_t \in \mathbb{R}^p$ and $h_t \in \mathbb{R}$ to provide a prediction value for current leaf. The training of oblique regression trees involves solving the following optimization problem:

$$\min_{\boldsymbol{A}, \boldsymbol{b}, \boldsymbol{K}, \boldsymbol{h}} \sum_{i=1}^n (y_i - \hat{y}_i)^2, \tag{1a}$$

$$\text{s.t. } \hat{y}_i = f_{tree}(\boldsymbol{A}, \boldsymbol{b}, \boldsymbol{K}, \boldsymbol{h}, \boldsymbol{x}_i), \ i \in \{1, \cdots, n\}, \tag{1b}$$

where $\boldsymbol{A} = \{\boldsymbol{a}_1, \cdots, \boldsymbol{a}_{\lfloor T/2 \rfloor}\}$ and $\boldsymbol{b} = \{b_1, \cdots, b_{\lfloor T/2 \rfloor}\}$ are tree split parameters for branch nodes, $\boldsymbol{K} = \{\boldsymbol{k}_{\lfloor T/2 \rfloor + 1}, \cdots, \boldsymbol{k}_T\}$ and $\boldsymbol{h} = \{h_{\lfloor T/2 \rfloor + 1}, \cdots, h_T\}$ are leaf prediction parameters. In this work, we consider two types of leaf prediction: **(a)** linear prediction and **(b)** constant prediction.

**(a) Tree with linear predictions:** Linear predictions involve a linear combination of input features (Quinlan, 1998), representing a general form of leaf predictions. If $\boldsymbol{x}_i$ is assigned to leaf node $t$, the final prediction is described as $\hat{y}_i = \boldsymbol{k}_t^T \boldsymbol{x}_i + h_t$.

**(b) Tree with constant predictions:** This type is a special case of linear predictions, where $\boldsymbol{K}$ remains zero. It is the most commonly used type in existing decision tree methods, with $\hat{y}_i = h_t$.

## 3 Unconstrained Optimization Formulation

In this work, we reformulate the tree training as an unconstrained optimization task, allowing us to leverage powerful gradient-based optimization frameworks for improved solvability and accuracy.

### 3.1 Deterministic Sample Route Formulation

The interpretability of decision trees primarily stems from their transparent prediction rules associated with the tree paths from the root to leaf nodes. Identifying a sample's specific tree path, such as "$1 \to 2 \to 5$" for a sample routed to leaf node 5, is crucial for calculating prediction loss.

Specifically, for a leaf node $t \in \mathbb{T}_L$, we denote its set of ancestor nodes as $\mathbb{A}_t$. The subsets $\mathbb{A}_t^l$ and $\mathbb{A}_t^r$ represent the ancestor nodes traversed via the left branch and right branch, respectively, such that $\mathbb{A}_t = \mathbb{A}_t^l \cup \mathbb{A}_t^r$. Additionally, we introduce a binary branching test variable $I_{i,j} \in \{0,1\}$ to signify whether a sample $\boldsymbol{x}_i$ successfully passes the branching test at the branch node $j$, defined as $I_{i,j} = \mathbb{1}\left(b_j - \boldsymbol{a}_j^T \boldsymbol{x}_i > 0\right)$. Here, $I_{i,j} = 1$ signifies a successful pass at node $j$; otherwise, $I_{i,j} = 0$.

Subsequently, the sample routing indicator $P_{i,t} \in \{0,1\}$, determines if sample $\boldsymbol{x}_i$ is assigned to leaf node $t$, computed as follows:

$$P_{i,t} = \prod_{j \in \mathbb{A}_t^l} I_{i,j} \prod_{j \in \mathbb{A}_t^r} (1 - I_{i,j}), \tag{2}$$

where $P_{i,t} = 1$ indicates an assignment to the leaf node $t$, and $P_{i,t} = 0$ denotes non-assignment. An illustrative example for understanding these formulations is provided in Appendix A, Figure 4.

## 3.2 Loss Formulation and Differentiability

Following the deterministic sample route, the training of oblique regression trees is subsequently reformulated as an unconstrained optimization problem. The objective function $\mathcal{L}$ is defined by

$$\mathcal{L} = \sum_{i=1}^{n} \sum_{t \in \mathbb{T}_L} P_{i,t} \left( y_i - \left( \boldsymbol{k}_t^T \boldsymbol{x}_i + h_t \right) \right)^2. \tag{3}$$

Here, the variables $\boldsymbol{A}$, $\boldsymbol{b}$, $\boldsymbol{K}$ and $\boldsymbol{h}$ are implicitly expressed in terms of $I_{i,j}$ and $P_{i,t}$, as shown in Equation (2). For decision trees with constant predictions, the variable $\boldsymbol{K}$ is always equal to zero.

In the computation of gradients of $\mathcal{L}$ (as detailed in Appendix B, Figure 5), an exception arises due to the non-differentiability of the indicator function $\mathbb{1}(\cdot)$ in the calculation of the branching test variable $I_{i,j}$. To resolve this issue, we employ the scaled sigmoid function $\mathbb{S}(\cdot)$ as an approximation for the indicator function, resulting in the introduction of the approximated branching test variable denoted as $\hat{I}_{i,j}$:

$$\hat{I}_{i,j} = \mathbb{S}(b_j - \mathbf{a}_j^T \mathbf{x}_i) = \left[ 1 + e^{-\alpha \left( b_j - \mathbf{a}_j^T \mathbf{x}_i \right)} \right]^{-1}, \tag{4}$$

where $\alpha$ represents a critical balance between achieving high approximation accuracy and maintaining stability in optimization processes, while also providing some potential concerns.

**Concerns regarding the special case:** $\alpha = 1$ corresponds to standard sigmoid function, commonly used in Soft Decision Tree (İrsoy et al., 2012; Frosst & Hinton, 2017), which adopt soft splits at branch nodes and probabilistic predictions at leaf nodes. Their work on soft trees deviates from the interpretability pertinent to hard True-False decision and deterministic predictions, making it less suitable for scenarios requiring hard-splits, as exemplified in Appendix C. More importantly, there remains a big gap between the standard sigmoid function with $\alpha = 1$ and the true indicator function, diminishing the accuracy of the training.

**Concerns regarding the selection of $\alpha$:** A larger $\alpha$ leads to a more accurate approximation, but it also introduces numerical instability, potentially compromising the optimization capabilities. Further empirical analyses are given in Appendix D. Identifying the optimal $\alpha$ that balances approximation degree and differentiability remains a challenge. To mitigate this, we propose an iterative scaled sigmoid approximation strategy, detailed in the following Section 4.1, to narrow the gap between the original indicator function and its differentiable approximation.

**Clarification for the adoption of hard-split in inference:** Unlike soft trees that base final predictions on probability and trained leaf values ($\boldsymbol{K}$ and $\boldsymbol{h}$), our tree still maintains hard-splits. Hard splits in the inference phase not only meet the practical demand for hard decisions but also mitigates additional errors that may arise from the trained leaf values. Ideally, with a high approximation accuracy and optimal optimization, the trained leaf values from soft approximations should closely match those calculated by hard splits. However, achieving this level of optimality is challenging due to potential errors in soft approximation. This concern justifies the use of hard splits in the inference phase, which are more likely to yield more accurate final predictions.

## 4 Oblique Tree Training Through Gradient-based Optimization

Our optimization task in Equation (3), incorporating the approximated $\hat{I}_{i,j}$ obtained from Equation (4), closely approximate the original non-differentiable tree training problem. This task can be efficiently solved through our proposed entire tree optimization framework.

### 4.1 Iterative Scaled Sigmoid Approximation

In response to early-mentioned concerns regarding an optimal $\alpha$, we propose a strategy of iterative scaled sigmoid approximation to enhance the approximation accuracy to indicator functions. The key challenge of lies in the selection of $\alpha$. A larger $\alpha$ may destabilize optimization process, whereas a smaller $\alpha$ tends to be easier to solve for gradient-based optimization. Our strategy leverage this insight by using a solution from an optimization task with a smaller scale factor to effectively warm-starts optimization with a larger scale factor. By starting with a smaller scale factor and gradually

increasing it, this strategy enhance the approximation accuracy, while mitigating numerical instability typically associated with larger scale factors.

Specifically, the procedure begins by randomly sampling a set of scale factors within a predetermined range, ensuring a broad exploration of possible $\alpha$ values. These sampled scale factors, denoted as $\{\alpha_1, \cdots, \alpha_n\}$, are then applied in ascending order, from small to large. We initiate the optimization with the smallest sampled scale factor to generate the initial optimized tree candidate. This candidate then serves as the starting point for the subsequent optimization task with a slightly larger $\alpha$. This iterative process is repeated until all sampled scale factors have been utilized. Detailed implementation steps are integrated within our systematic optimization framework, Algorithm 1.

## 4.2 Gradient-based Entire Tree Optimization Framework

Unlike greedy methods that optimize each node sequentially, our approach concurrently optimizes the entire tree, encompassing tree split parameters $\boldsymbol{A}$ and $\boldsymbol{b}$ at all branch nondes, and leaf prediction parameters $\boldsymbol{K}$ and $\boldsymbol{h}$ at all leaf nodes. Our entire tree optimization, outlined in Algorithm 1, begins at multiple random initialization (*Line 4 - 6*). This multiple-initialization serves two purposes: First, multi-start increases the chance of finding the optimal solution of the unconstrained reformulation. Second, in Section 4.1, the iterative scaled sigmoid approximation involves randomly sampling scale factors for each iteration. These multiple starts enable sampling diverse scale factors in a wider range, thereby enhancing the robustness and approximation accuracy. For each start, the optimization with iterative scaled sigmoid approximation is implemented to produce an optimized tree candidate (*Line 7 - 17*). Importantly, our method deterministically calculates the leaf values based on hard-splits (*Line 12*). The specific deterministic calculations for $\boldsymbol{K}$ and $\boldsymbol{h}$ are given in Appendix E. Finally, the optimal tree is determined by comparing each candidate from multiple starts (*Line 14 - 15*). This optimization framework is readily implementable using existing powerful tools that embed gradient-based optimizers, such as PyTorch and TensorFlow.

The framework is applicable to decision trees with both constant and linear predictions. The only minor difference is that for constant predictions, the parameters $\boldsymbol{K}$ remain zero without gradients. For clarity, we term our approach **G**radient-based **E**ntire **T**ree optimization as `GET` when applied to trees with constant predictions; otherwise, termed as `GET-Linear` for trees with linear predictions.

---

**Algorithm 1** Gradient-based Entire Tree Optimization (applicable to both `GET` and `GET-Linear`)

1: **Input:** $\{\boldsymbol{x}_i, y_i\}_{i=1}^n$, tree depth $D$, learning rate $\eta$, epoch number $N_{epoch}$, multi-start number $N_{start}$.
2: **Output:** Optimal trainable variables $\boldsymbol{A}_{best}$, $\boldsymbol{b}_{best}$, $\boldsymbol{K}_{best}$ (Zero for the case of `GET`) and $\boldsymbol{h}_{best}$.
3: Assign a large value to the parameter $\mathcal{L}_{min}$ and define empty variables for $\boldsymbol{A}_{best}$, $\boldsymbol{b}_{best}$, $\boldsymbol{K}_{best}$ and $\boldsymbol{h}_{best}$.
4: **for** $start = 1$ **to** $N_{start}$ **do**
5:     Initialize trainable variables $\boldsymbol{A}$, $\boldsymbol{b}$, $\boldsymbol{K}$ (Zero for `GET`) and $\boldsymbol{h}$.
6:     Randomly generate a set of scale factors $\{\alpha_1, \cdots, \alpha_n\}$ in ascending order.
7:     **for** $\alpha_{iter} \in \{\alpha_1, \cdots, \alpha_n\}$ **do**
8:         If $iter \neq 1$, initialize trainable variables from the solution of last iteration computed by *Line 13*.
9:         **for** $k = 1$ **to** $N_{epoch}$ **do**
10:             Approximate loss $\mathcal{L}$ at step $k$ by Equation (3) and (4) and calculate $\frac{\partial \mathcal{L}}{\partial \boldsymbol{A}}$, $\frac{\partial \mathcal{L}}{\partial \boldsymbol{b}}$, $\frac{\partial \mathcal{L}}{\partial \boldsymbol{K}}$ (exclude for `GET`) and $\frac{\partial \mathcal{L}}{\partial \boldsymbol{h}}$. Then update trainable variables, such as $\boldsymbol{A}_{k+1} = \boldsymbol{A}_k - \eta \frac{\partial \mathcal{L}}{\partial \boldsymbol{A}}$.
11:         **end for**
12:         Deterministically update $\boldsymbol{K}$ and $\boldsymbol{h}$ based on hard-splits.
13:         Generate a tree candidate with optimized variables, termed as $\boldsymbol{A}_{iter}$, $\boldsymbol{b}_{iter}$, $\boldsymbol{K}_{iter}$ and $\boldsymbol{h}_{iter}$.
14:         Deterministically compute the current loss $\mathcal{L}$ based on hard-splits using Equation (3).
15:         IF $\mathcal{L} < \mathcal{L}_{min}$, $\boldsymbol{A}_{best} \leftarrow \boldsymbol{A}_{iter}$; $\boldsymbol{b}_{best} \leftarrow \boldsymbol{b}_{iter}$; $\boldsymbol{K}_{best} \leftarrow \boldsymbol{K}_{iter}$; $\boldsymbol{h}_{best} \leftarrow \boldsymbol{h}_{iter}$; $\mathcal{L}_{min} \leftarrow \mathcal{L}$.
16:     **end for**
17: **end for**

---

**Hyperparameters Analysis:** Despite the introduction of additional hyperparameters in gradient-based optimization, tuning them is not typically necessary because their effects are straightforward. For instance, the multi-start number $N_{start}$ directly influences training optimality by increasing the chance of finding optimal solution, albeit at a higher computational cost. In practice, $N_{start}$ is set to balance acceptable computational cost with desired training accuracy. More empirical analyses of $N_{start}$, the sampling range of scale factors and other hyperparameters are detailed in Appendix F.

### 4.3 Subtree Polish Strategy to Mitigate Accumulated Approximation Errors

**Accumulated Approximation Error Analysis:** Despite our approximation strategy effectively narrowing the substantial gap to the indicator function at each node, the approximation error can still accumulate across an entire decision tree, particularly in deeper trees with numerous nodes. This accumulation can significantly degrade the training optimality, a concern that has received insufficient attention in the literature. To mitigate these accumulated errors, we propose a subtree polish strategy to further enhance training optimality.

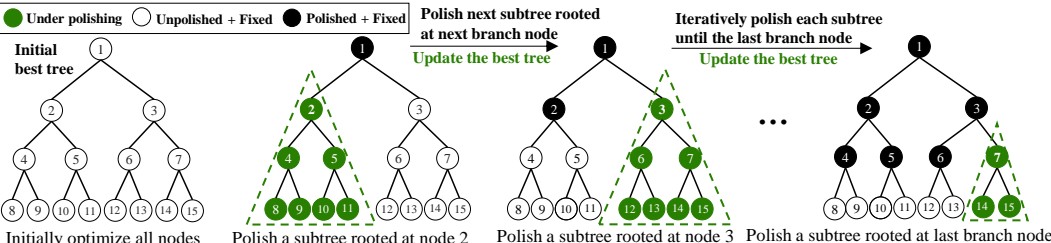

Figure 1: The illustrative example of our subtree polish strategy.

**Subtree Polish Strategy to improve training optimality:** In Algorithm 1, Our method is designed to simultaneously optimize all $2^D - 1$ branch nodes of an entire tree. Intuitively, once a branch node is optimally identified, we can sequentially polish its subtree, including all child nodes of that branch node, while keeping the rest of the tree fixed. This process forms the basis of our subtree polish strategy, which starts with optimizing the entire tree, establishing an initial best tree candidate.

For each branch node, there exists a corresponding subtree that extends from the current branch node to the leaf nodes. As illustrated in Figure 1, the subtree is represented within a dashed triangle. Each subtree is optimized using the same approach as for entire tree in Algorithm 1, while leaving the remaining tree nodes fixed. Each subtree optimization is warm-started with the best tree candidate available at that time. Once a subtree is optimized, we combine it with the fixed tree nodes and update the best tree if the combined tree improves training accuracy. This process then proceeds to next subtree, rooted at next branch node. This iterative process continues until the subtree rooted at the last branch node is polished. The final best tree is returned after polishing all branch nodes. The implementation procedure of the subtree polish strategy is provided in Appendix G, Algorithm 2.

## 5 Numerical Experiments and Discussions

We evaluate our optimized tree with both constant and linear predictions, termed as `GET` and `GET-Linear`, against random forests `RF`. Our analysis covers testing accuracy, the number of parameters and prediction time. Additionally, we assess the capabilities of `GET`, with other decision tree methods in terms of both training optimality and testing accuracy. Finally, we provide a limitation analysis. The compared tree methods include the baseline `CART`, greedy methods like `HHCART` (Wickramarachchi et al., 2015), `RandCART` (Blaser & Fryzlewicz, 2016), and `OC1` (Murthy et al., 1994), existing gradient-based trees such as `GradTree` (Marton et al., 2023) using a straight-through estimator for non-differentiable splits, and soft decision tree `SoftDT` (Frosst & Hinton, 2017) using standard sigmoid function for soft approximation, as well as the state-of-the-art heuristic method `ORT-LS` (Dunn, 2018). These comprehensive experiments are conducted on 16 real-world datasets obtained from UCI machine learning repository (Dua & Graff, 2019) and OpenML (Vanschoren et al., 2014), with sample size ranging from 1,503 to 16,599 and feature number from 4 to 40. Detailed dataset information and specific data usage in our study is provided in Appendix H.1. Comparisons focus on testing and training accuracy in terms of $R^2$, and computational time in seconds. The Friedman Rank (Sheskin, 2020) is also used to statistically sort the compared methods according to their testing accuracy, with a lower rank indicating better performance. Comprehensive details on the implementation, algorithm configuration, and computing facilities are provided in Appendix H.2.

## 5.1 TESTING ACCURACY COMPARISON AGAINST RANDOM FORESTS

For testing accuracy comparison, we conduct depth tuning by cross validation to determine the optimal depth across depths from 1 to 12 for our methods. For a fair comparison, comprehensive hyperparameter tuning is also performed for RF. Specifically, the number of trees in a forest is a critical parameter. It is well-recognized that testing performance improves with an increase in the number of trees; however, the marginal gains become less pronounced as additional trees are added (Probst & Boulesteix, 2018; Oshiro et al., 2012). Accordingly, the number of trees for RF is tuned across a set of $\{50, 100, 200, 300, 400, 500\}$. Moreover, the maximum tree depth for RF is tuned over a broader range, from 1 to 50, to potentially capture optimal depth settings, given that RF empirically benefits from overly-deeper trees for enhanced testing accuracy. Other hyperparameters for RF, including the number of features per split and the number of samples per tree, are maintained at default settings. These parameters have been shown to balance the bias-variance trade-off, typically yielding robust performance with default values (Probst & Boulesteix, 2018).

**Testing accuracy comparison:** Our GET slightly underperforms RF by 0.24% averaging across 16 datasets. In contrast, GET-Linear significantly outperforms RF by 2.03%, as detailed in Table 1. This superiority is further supported by the Friedman Rank, where GET-Linear ranking the highest, followed by RF and GET. Among 16 datasets, GET-Linear outperforms RF in 8 datasets. Detailed results for each dataset are given in Appendix H.3. These findings question the conventional belief that a single decision tree typically underperforms random forests, highlighting the capabilities of our approach in achieving competitive testing accuracy.

Table 1: Comparison of testing accuracy for GET, GET-Linear, and RF across 16 datasets.

| Item | Number of Trees | Tree Depth | Test Accuracy (%) | Friedman Rank |
|---|---|---|---|---|
| RF | 309.38 | 19.24 | 81.94 | 1.75 |
| GET | 1 | 6.56 | 81.70 | 2.69 |
| GET-Linear | 1 | 6.94 | **83.97** | 1.56 |

**Paired T-test for statistical significance:** While Table 1 empirically shows that our single tree is competitive to RF, we further conduct a Paired T-test to statistically validate the significant difference among GET, GET-Linear and RF. The null hypothesis asserts that RF is not significantly different from GET and GET-Linear. By setting a tolerance (acceptable significance level) $\tau = 0.1$, if calculated $p$-value is less than $\tau$, the null hypothesis can be rejected, indicating statistical significance.

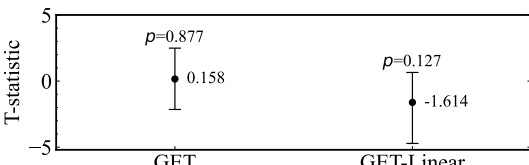

Figure 2: T-statistic and p-value of the Paired T-test comparing RF with our methods.

The t-statistic (black point), p-value, 95% confidence interval are depicted in Figure 2. The Paired T-test between RF and our GET yields a $p$ value of 0.877 (greater than $\tau$), suggesting no statistically significant difference in testing accuracy between RF and GET. Thus, their testing accuracies are comparable. In contrast, the $p$ value for T-test between RF and GET-Linear is relatively smaller with $p = 0.127$, also suggesting that RF is not significantly better than GET-Linear. Moreover, if we accept a tolerance $\tau > 0.127$, we can reject the null hypothesis of equal performance. As indicated by a negative t-statistic of -1.164, this implies that GET-Linear is statistically superior to RF. These results further confirm the competitive testing accuracy of our tree over random forests.

## 5.2 TESTING ACCURACY COMPARISON AGAINST OTHER DECISION TREES

Following our empirical findings, which showcases the superiority of our optimized tree, we proceed to compare it with other commonly-used decision tree methods under same depth tuning setting, to further validate the efficacy of our approach. For a fair comparison, since existing decision trees are mainly designed with constant predictions, we limit the comparison to our GET with them.

Table 2: Comparison of test accuracy for GET and other decision tree methods.

| Item | Greedy Methods | | | | Gradient-based Trees | | State-of-the-Art Heuristic | Our Tree |
|---|---|---|---|---|---|---|---|---|
| | CART | OC1 | RandCART | HHCART | SoftDT | GradTree | ORT-LS | GET |
| Test Accuracy (%) | 74.18 | 72.54 | 71.31 | 76.37 | 72.69 | 64.13 | 78.01 | **81.77** |
| Tree Depth | 10.06 | 7.94 | 8 | 8.19 | 10.19 | 10.38 | 6.13 | 6.56 |
| Friedman Rank | 4.81 | 5.19 | 5.88 | 3.63 | 4.69 | 6.94 | 3.50 | **1.38** |

**Testing accuracy comparison:** Our GET consistently outperform compared decision trees in testing accuracy across 16 datasets, as shown in Table 2. Specifically, GET achieves the highest testing accuracy, surpassing the state-of-art heuristic method ORT-LS by 3.76%, the greedy method HHCART by 5.39%, and the baseline orthogonal tree CART by 7.59%. Notably, compared to other gradient-based trees, GET also outperforms GradTree by 17.64%, and the soft decision tree SoftDT by 9.08%. These results underscore the effectiveness of our tree optimization approach. Additionally, the rank comparisons reinforce these findings, with GET attaining the highest rank. Detailed results for each dataset are provided in Appendix H.3.

**Paired T-test for statistical significance:** The Paired T-test for all comparisons consistently shows that we can reject the null hypothesis, which posits that our method GET is not significantly different from the compared decision trees. The observations of $p < \tau = 0.1$ and positive t-statistic value, indicate that GET is superior to these compared decision trees with statistically significance.

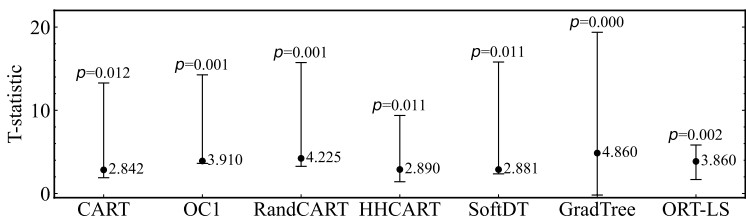

Figure 3: The Paired T-test comparing our GET with various decision tree methods.

## 5.3 SUPERIOR TEST ACCURACY ANALYSIS: FROM TRAINING OPTIMALITY PERSPECTIVE

To figure out the rationale behind the competitive testing accuracy of our optimized tree, we delve into the analysis of training optimality from optimization perspective. Notably, the training of tree with different depth corresponds to different optimization problems, exhibiting different optimization challenge in scalability and performance. To assess the optimization capabilities, the training of a predetermined-depth tree is a common practice in the literature. Therefore, the predetermined depths of $D = \{2, 4, 8, 12\}$ are used for training accuracy comparison between these top four methods reported in Table 2. Detailed comparisons for all methods are given in Appendix H.4.

Table 3: Training accuracy comparison on 16 real-world datasets across different depths.

| Depth | Training Accuracy (%) | | | | Testing Accuracy(%) | | | |
|---|---|---|---|---|---|---|---|---|
| | CART | HHCART | ORT-LS | GET | CART | HHCART | ORT-LS | GET |
| 2 | 47.26 | 46.68 | 66.47 | 71.80 | 46.45 | 46.12 | 64.44 | 70.24 |
| 4 | 60.90 | 62.59 | 79.51 | 82.40 | 58.60 | 61.24 | 74.84 | 77.89 |
| 8 | 81.33 | 82.28 | 90.81 | 91.02 | 69.07 | 74.18 | 74.00 | 78.55 |
| 12 | 93.43 | 94.90 | 97.50 | 96.09 | 67.21 | 67.63 | 67.14 | 71.97 |

**The effectiveness of our tree optimization approach:** Table 3 shows that our method GET outperforms ORT-LS by 5.33%, 2.89% and 0.21% in training accuracy for depths of 2, 4 and 8, respectively, while it outperforms CART by 24.55%, 21.50% and 9.69% across various depths. These improvements in training accuracy verify the efficacy of our tree optimization method to achieve training optimality. Additionally, an increase in training accuracy correlates with improved testing accuracy at depths of 2, 4, and 8, suggesting that an optimized tree with higher training accuracy can potentially yield better testing accuracy before encountering serious overfitting issues. Overfitting, particularly at deeper depth like 12, is simply addressed by tuning an optimal tree depth through cross validation, as discussed in previous subsections.

**Ablation study on the strategies of our optimization approach:** The observed improvements can be attributed to two key strategies: the iterative scaled sigmoid approximation, which improves training accuracy by an average of 9.95% across depths of 2, 4, 8, and 12 compared to the simple sigmoid approximation; and the subtree polish strategy, which contributes an additional average improvement of 1.4%. Detailed results for the ablation study are provided in Appendix H.5.

**Training Time Analysis:** A detailed training time comparison is given in Appendix H.4. Our method `GET` not only significantly improves training accuracy but also shows great scalability over the state-of-the-art `ORT-LS` by approximately 20 times acceleration at deep depth 12. However, it remains considerably slower, thousands of times, than `CART`. This trade-off between training optimality and computational cost is deemed acceptable for a single tree model. In comparison with random forests, our `GET` offers competitive accuracy but incurs substantially longer training times, being thousands of times slower than `RF`. Despite this inefficiency, our focus mainly lies in making a single tree suitable for scenarios with limited computational resources. Consequently, our primary metrics of interest are testing accuracy, interpretability and testing time, rather than training time. The benefits of its lightweight structure and prediction speed are further discussed in Section 5.4.

## 5.4 ANALYSIS FOR PARAMETER NUMBER, PREDICTION TIME, AND INTERPRETABILITY

As discussed in Section 1, random forests usually replace a single decision tree to improve testing accuracy in embedded systems. However, a single tree with comparable accuracy could be preferable due to its lightweight structure and interpretability. In this section, we primarily compare our methods with `RF` regarding the number of parameters, prediction time, and interpretability.

**Comparison of parameter number and testing time:** Building on the testing comparison in Table 1, we then assess total parameter number and prediction time, in Table 4. `RF` contains 324 times more parameters than `GET` and 119 times more than `GET-Linear`. The prediction speed of `GET`, averaged over 10,000 repetitions, is 30 times faster than `RF`, and `GET-Linear` is 24 times faster. This comparison shows that our tree achieves competitive accuracy with significantly fewer parameters and faster prediction than `RF`, thereby saving memory and computational costs.

Table 4: Comparison of parameter number and prediction time for `GET`, `GET-Linear`, and `RF`.

| Item | Number of Branch Nodes | Number of Leaf Nodes | Parameter Number in Branch Nodes | Parameter Number in Leaf Nodes | Total Parameters | Prediction Time (s) |
|---|---|---|---|---|---|---|
| RF | 819,566.56 | 819,875.94 | 1,639,133.13 | 819,875.94 | 2,459,009.06 | 1.7337 |
| GET | 476.50 | 477.50 | 7,101.63 | 477.50 | 7,579.13 | 0.0572 |
| GET-Linear | 1,084.25 | 1,085.25 | 10,292.13 | 10,304.50 | 20,596.63 | 0.0728 |

**Interpretability of our oblique tree:** Interpretability can be assessed from two aspects: tree-based prediction logic and the complexity of decision rules. First, `RF`, in Table 1, uses an average of 309.38 trees for higher accuracy, almost losing the interpretability for final predictions compared to a single tree. Second, `RF` often results in an overly-deep tree with an average depth of 19.24, significantly deeper than our `GET` with depth of 6.56. Moreover, as compared in Table 2, `GET` not only achieves the highest testing accuracy but also with the lowest tree depth, enhancing interpretability. For instance, a 2-depth tree yields 4 decision rules across 2 layers, whereas 8-depth tree produces 256 rules across 8 layers. Understanding hundreds of nested IF-THEN rules can be challenging. Therefore, our optimized tree with smaller depth offers more interpretability than deeper decision trees.

## 5.5 LIMITATIONS ANALYSIS OF OUR APPROACH

Our extensive experiments are conducted on datasets containing less than 20,000 samples, without exploring large-scale datasets. While our approach exhibits great scalability compared to the state-of-the-art heuristic decision tree `ORT-LS`, it is still thousands of times slower than the heuristic methods `CART` and `RF`. This limitation is generally acceptable for a single tree model, particularly because it significantly improves accuracy, aligning with our primary focus on testing accuracy, interpretability, and prediction time. However, the longer training time could potentially limit the application of our method in scenarios involving large-scale datasets.

Another limitation arises from inadequate regularization in our optimization approach. As shown in Table 3, while training accuracy for `GET` improves by 5.07% from depth 8 to 12, testing accu-

racy conversely drops by 6.58%, indicating a serious overfitting issue. This overfitting issue is more pronounced in decision trees with linear predictions, with more detailed analysis and comparisons provided in Appendix H.6. In response, we preliminarily apply $L_1$ regularization to `GET-Linear` for the experiments reported in Table 1, leading to a modest improvement in testing accuracy by 0.73%. However, due to the challenges in identifying the optimal regularization strength and potential increases in computational costs, we limited our tuning to only between 0 and a small value of $1e - 5$, without extensive tuning. Despite these efforts, further enhancements in testing accuracy could be achieved through more dedicated regularization strategies.

## 6 CONCLUSION

In conclusion, our development of the gradient-based entire tree optimization method, is not necessarily to bring the best regression tree, but rather to explore the potential of a single decision tree in achieving comparable testing accuracy to random forests. This makes a single tree more preferable for scenarios where a lightweight structure and interpretability are valued alongside predictive performance. Our approach reformulates decision tree training as a differentiable unconstrained optimization task, incorporating an iterative scaled sigmoid approximation. The tree optimization capability is further enhanced by a subtree polish strategy. Extensive experiments show that our optimized tree not only achieves but also statistically confirms its testing accuracy comparable to random forest, challenging the mindset that a single decision tree typically underperforms random forests in testing performance.

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

## A    EXAMPLE OF DETERMINISTIC SAMPLE ROUTING FORMULATIONS

We introduced the deterministic sample routing formulations in Section 3.1. To facilitate a better understanding of these formulations, an illustrative example is provided. As shown in Figure 4, we use a 2-depth tree, where the leaf nodes are indexed by $\mathbb{T}_L = 4, 5, 6, 7$. Focusing on leaf node $t = 5$, the corresponding tree path is delineated as "$1 \rightarrow 2 \rightarrow 5$". The associated ancestor sets are specified as $\mathbb{A}_5^l = 1$ and $\mathbb{A}_5^r = 2$. Assuming the $i$th sample is assigned to leaf node 5 by successfully passing the branching test on branch node 1 with $\boldsymbol{a}_1^T \boldsymbol{x}_i < b_1$ and failing on branch node 2 with $\boldsymbol{a}_2^T \boldsymbol{x}_i \geq b_2$, the ensuing outcomes are $I_{i,1} = 1$, $I_{i,2} = 0$, $P_{i,5} = I_{i,1} \cdot (1 - I_{i,2}) = 1$, and $P_{i,4} = P_{i,6} = P_{i,7} = 0$. Consequently, this signifies that the sample $\boldsymbol{x}_i$ is assigned to leaf node 5.

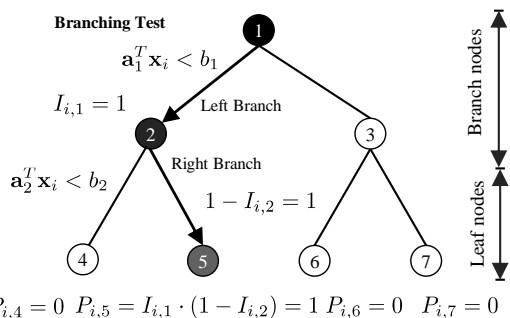

Figure 4: Example of deterministic sample routing $1 \rightarrow 2 \rightarrow 5$.

## B    ILLUSTRATION OF FORWARD AND BACKWARD PROCESSES OF LOSS CALCULATION.

Along with detailing gradient calculation for the loss function in Section 3.2, we also provide a comprehensive illustration of the forward and backward processes involved in loss and gradient calculation.

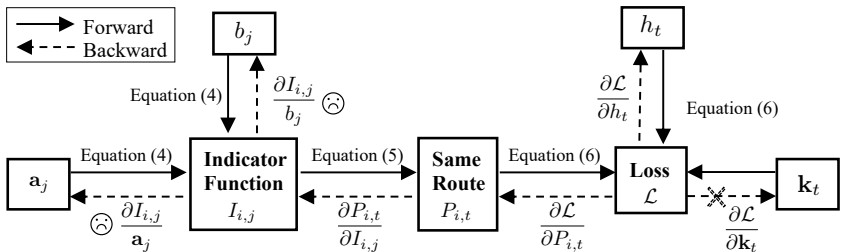

Figure 5: Forward and backward processes of loss calculation.

## C    CERTAIN SCENARIOS REQUIRING DECISION TREES WITH HARD-SPLITS

Hard-split decision trees and soft decision trees represent fundamentally different models, each suited to different application scenarios. Further, there do exist scenarios where the application of hard-split decision trees is not only more appropriate but also imperative. Below are illustrative examples from several industrial projects that underscore this preference:

Scenario 1: Threshold-based well control optimization in underground hydrogen storage system.

In the context of large-scale underground hydrogen storage, the operation of periodic hydrogen injection and production across numerous wells necessitates the design of optimal control strategies. Such decision-making problems can be modeled by decision trees, particularly when decisions center on exceeding specific thresholds. For instance, the reservoir pressure or the production rate

exceeding a certain threshold can be regarded as the hard decisions at branch nodes. For these scenarios, hard-split decision trees are more suitable for making clear decisions, which can be easily understood and implemented by field engineers or operators.

Scenario 2: True-false decision-making in law enforcement.

In law enforcement, decision-making often involves binary choices, such as whether to prosecute a suspect or not. In this case, hard-split decision trees can be used to make clear decisions based on the evidence and the law, which can be easily interpreted by legal professionals.

Scenario 3: Piece-wise affine control law in explicit model predictive control.

In explicit model predictive control, the control law is often represented by piece-wise affine functions, which can be effectively approximated by a hard-split decision tree. The hard decisions at branch nodes can be used to determine the control action based on the state of the system, which can be easily implemented in real-time control systems.

Beyond the utility of making clear hard decisions, the optimal decision tree model that offers superior predictive performance is also crucial for these scenarios. From the perspective of application scenarios, the motivation and necessity for hard-split optimal decision trees are both intuitive and compelling.

## D   EMPIRICAL ANALYSIS OF THE IMPACT OF SCALE FACTORS ON SCALED SIGMOID APPROXIMATION

The scale factor $\alpha$ significantly influence the approximation degree to indicator function and the behavior of its gradient in optimization. A larger $\alpha$ leads to a better approximation than sigmoid function, achieving closer proximity to an indicator function as $\alpha$ approaches infinity. Nonetheless, a larger $\alpha$ also results in a more unstable gradient, which may adversely affect the optimization process. Specifically, a larger $\alpha$ may cause the gradient to be too small or even zero, leading to the stagnation of gradient descent.

The gradient of scaled sigmoid function, denoted as $S(x) = (1 + e^{-\alpha x})^{-1}$ is given by $\frac{\partial S}{\partial x} = \alpha S(x)(1 - S(x))$. When $\alpha$ is larger, either $S(x)$ or $1 - S(x)$ tend towards zero for value of $x$ away from origin, potentially causing the gradient to approach zero. Consequently, the $\alpha$ represents a critical balance between achieving high approximation degree and maintaining stability in optimization processes.

To show how the scale factor $\alpha$ impacts the training optimality of the unconstrained optimization problem (our method without using the strategy of iterative scaled sigmoid approximation), we conduct the comparison experiments across different $\alpha$ values at different depths. The findings, summarized in Table 5, reveal the relationship between $\alpha$ and training performance, as measured by the average training accuracy $R^2$. Notably, we observe that the training accuracy at the extremes of $\alpha = 1$ (sigmoid function) and $\alpha = 1000$ are inferior compared to intermediate $\alpha$ values. This observation underscores two critical insights: firstly, relying solely on the sigmoid function ($\alpha = 1$) yields suboptimal optimization results; secondly, the high $\alpha$ value may not necessarily lead to better optimality.

Table 5: The impact of $\alpha$ on training accuracy across different depths.

| Various $\alpha$ Value | Training Accuracy (%) | | | |
|---|---|---|---|---|
| | $D = 2$ | $D = 4$ | $D = 8$ | $D = 12$ |
| $\alpha = 1$ (Standard Sigmoid Function) | 55.23 | 65.38 | 81.91 | 93.42 |
| $\alpha = 50$ | 67.97 | 78.72 | 87.51 | 93.99 |
| $\alpha = 100$ | 68.48 | 77.22 | 87.09 | 94.11 |
| $\alpha = 1000$ | 65.37 | 75.64 | 83.26 | 93.42 |

However, it remains a challenge to identify the optimal scale factor $\alpha$ that balances the trade-off between approximation degree and differentiability. To mitigate this issue, we propose an iterative scaled sigmoid approximation strategy, detailed in the following Section 4.1, to narrow the gap between the original indicator function and its differentiable approximation.

# E    DETERMINISTIC CALCULATIONS FOR LEAF PREDICTION PARAMETERS

As outlined in Algorithm 1, our method deterministically calculates the leaf values (i.e., the values of $\boldsymbol{K}$ and $\boldsymbol{h}$), rather than directly using trained values for $\boldsymbol{K}$ and $\boldsymbol{h}$. Given a tree with tree split parameters $\boldsymbol{A}$ and $\boldsymbol{b}$, the deterministic tree path for each sample can be calculated by Equation (2), which allows for determining the total number of samples assigned to a specific leaf node $t \in \mathbb{T}_L$.

For decision trees with constant predictions, the value of $\boldsymbol{K}$ remains zero. The value of $\boldsymbol{h}$ at a leaf node $t$ is an average of true output values ($y_i$) of the samples assigned to that leaf node $t$.

For decision trees with linear predictions, the prediction at a leaf node is a linear combination of input features by fitting a linear correlation between all samples assigned to that leaf node. The leaf values at a leaf node $t$, $\boldsymbol{k}_t$ and $h_t$, are the linear coefficients determined by linear regression.

# F    HYPERPARAMETERS ANALYSIS OF OUR GRADIENT-BASED ENTIRE TREE OPTIMIZATION APPROACH

Despite the introduction of additional hyperparameters in gradient-based optimization, tuning them is not typically necessary because their effects are straightforward. To be more specific, the hyperparameters in our gradient-based entire tree optimization approach are as follows:

(1) The multi-start number $N_{start}$

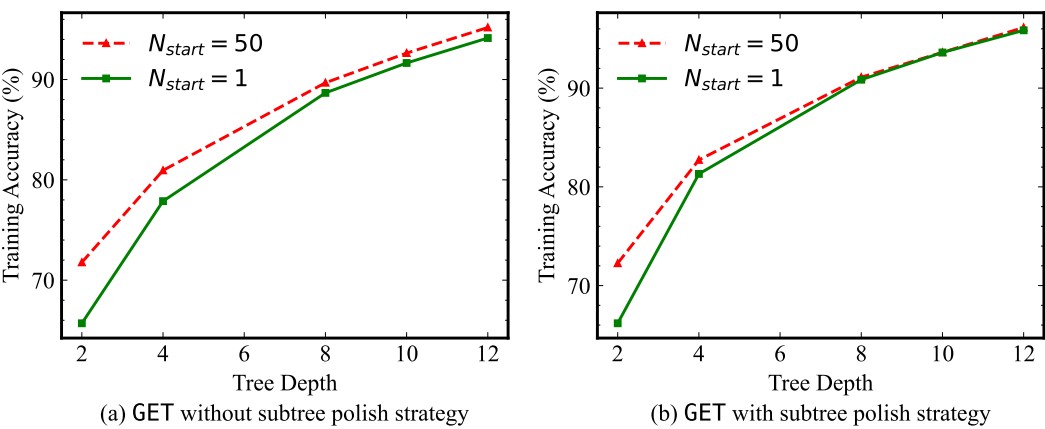

(a) GET without subtree polish strategy        (b) GET with subtree polish strategy

Figure 6: The trend of training optimality under different depth setting with different $N_{start}$.

The multi-start number $N_{start}$ directly influences training optimality by increasing the chance of finding the optimal solution, albeit at a higher computational cost. In practice, $N_{start}$ is set to balance acceptable computational expenses with desired training accuracy.

To explore the correlation between $N_{start}$ and the training optimality, our Gradient-based Entire Tree Optimization for decision trees with constant predictions (GET) is performed under different $N_{start}$ values as shown in Figure 6. It indicates that increasing $N_{start}$ generally improves training optimality for all various tree depths, especially at lower tree depths.

(2) The epoch number $N_{epoch}$

The epoch number $N_{epoch}$ is another hyperparameter that directly affects training optimality. A higher $N_{epoch}$ value increases training accuracy, but it also increases computational costs. In practice, $N_{epoch}$ is also set to balance acceptable computational expenses with desired training accuracy.

Our experiment with different $N_{epoch} = \{100, 3000, 5000\}$ in Figure 7, shows that increasing $N_{epoch}$ generally improves training optimality for all various tree depths. The improvements are more pronounced for lower $N_{epoch}$, and become less significant as $N_{epoch}$ increases.

(3) The range of sampled scale factors $[\alpha_{min}, \alpha_{max}]$

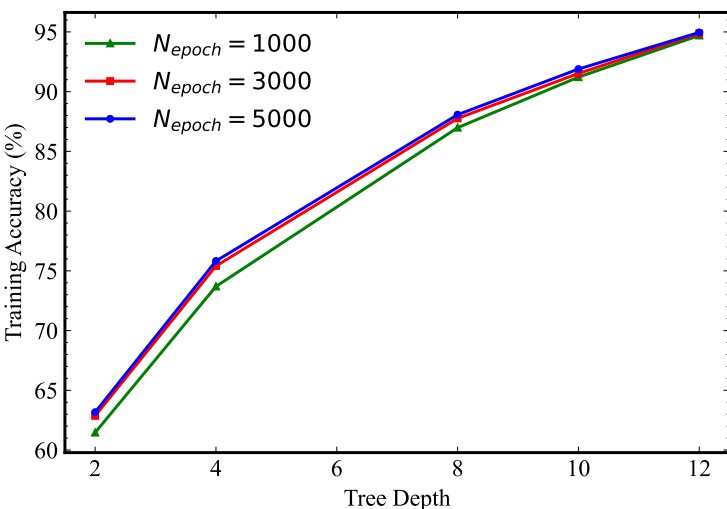

Figure 7: The trend of training optimality under different depth setting with various $N_{epoch}$.

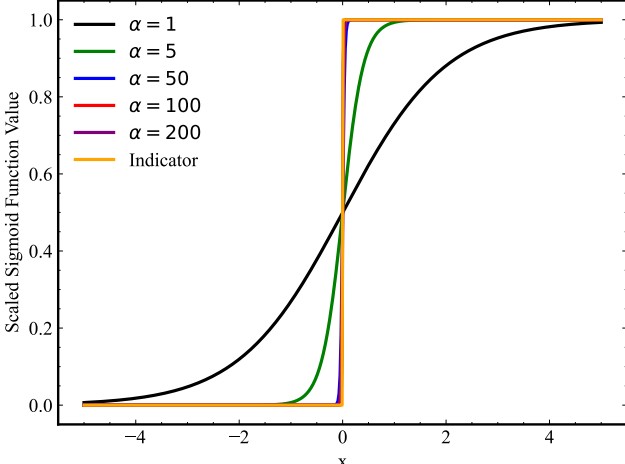

Figure 8: Disparity comparison of scaled sigmoid function over indicator function under varying $\alpha$.

This predetermined range is used to sample a set of scaled factors $\alpha$ for the strategy of iterative scaled sigmoid approximation. The principal aim is to explore a broader range of scale factors, ranging from smaller to larger values. As observed in Figure 8, the gap between scaled sigmoid approximation and the original indicator function narrows as $\alpha$ increases. Noticeably, the standard sigmoid function with $\alpha = 1$ exhibits a significant deviation from the indicator function as depicted in black line. In the implementation of our experiments, we simply set the range $[\alpha_{min}, \alpha_{max}] = [5, 150]$ meet our requirements. This range ensures that smaller values maintain a smooth gradient and exhibit less disparity than the standard sigmoid function, while larger values closely approximate the indicator function.

(4) The number of sampled scale factors

Within a predetermined range, a set of scaled factors, denoted as $\{\alpha_1, \cdots, \alpha_n\}$, is sampled and sorted in ascending order for subsequent use in iterative scaled sigmoid approximation, as detailed in Section 4.1. Larger scale factors reduce the approximation disparity, whereas smaller ones yield a smoother and more stable gradient. Including a greater number of scale factors in the set facilitates

a more stable approximation process, enhancing the approximation degree of the indicator function while minimizing the loss of differentiability typically associated with larger scale factors. Intuitively, including more scale factors in the set enhances training optimality. However, this leads to increased iterations in the approximation strategy, thereby raising computational costs. Practically, the number of scale factors is often determined by balancing training accuracy against computational demands. In our experiments, to avoid excessive computational costs, we primarily sample two scale factors: a smaller $\alpha$ within the range of $[5, 25]$ and larger $\alpha$ within the range of $[50, 150]$.

(5) The learning rate $\eta$

The learning rate is a common parameter in gradient-based optimization, and has garnered significant attentions in the literature. To simplify its usage, we adopt the well-established learning rate scheduler, referred to as `CosineAnnealingWarmRestarts` in PyTorch, which decreases the learning rate from an initial value of 0.01, thus minimizing the need for additional tuning.

## G  SUBTREE POLISH STRATEGY

---
**Algorithm 2** The Subtree Polish Strategy

---
1: **Input:** Dataset $\{\boldsymbol{x}_i, y_i\}_{i=1}^n$, tree depth $D$, and other parameters in Algorithm 1.
2: **Output:** Optimal trainable variables $\boldsymbol{A}_{best}$, $\boldsymbol{b}_{best}$, $\boldsymbol{K}$ (Zero for the case of `GET`) and $\boldsymbol{h}$.
3: Initially optimize an entire tree as the best tree candidate using Algorithm 1, termed $\boldsymbol{A}_{best}$ and $\boldsymbol{b}_{best}$.
4: **for** $t \in \mathbb{T}_B$ **do**
5:     Induce a subset $\{\mathcal{X}_t, \mathcal{Y}_t\}$ of the dataset $\{\boldsymbol{x}_i, y_i\}_{i=1}^n$ for branch node $t$ after fixing the optimized hyperplanes at its parent nodes.
6:     **if** $|\mathcal{X}_t| > 1$ and unique$(\mathcal{Y}_t) > 1$ **then**
7:         Retrieve $d_{sub}$-depth subtree results rooted at node $t$ from the tree candidate as the warm start.
8:         Polish the subtree, termed as $\boldsymbol{A}_{sub}$ and $\boldsymbol{b}_{sub}$, using Algorithm 1.
9:         Replace the corresponding subtree of the current best tree candidate with $\boldsymbol{A}_{sub}$ and $\boldsymbol{b}_{sub}$ from *Line 8*. Update the current best tree candidate only if this modification improves training optimality.
10:     **end if**
11: **end for**
12: Deterministically calculate $\boldsymbol{K}$ and $\boldsymbol{h}$ based on the final best tree candidate $\boldsymbol{A}_{best}$ and $\boldsymbol{b}_{best}$.

---

## H  THE BASIC SETTING OF NUMERICAL EXPERIMENTS

### H.1  DATASET INFORMATION

The 16 real-world dataset from UCI repository (Dua & Graff, 2019) and OpenML (Vanschoren et al., 2014) are used in our numerical experiments. Detailed information about these datasets is summarized in Table 6. The dataset size $n$ and the number of features $p$ are provided in the table.

Typically, we allocated 75% of the samples for training purposes and the remaining 25% for testing. If an experiment requires cross validation for hyperparameters tuning like tree depth, we then subdivide the training datasets into training and validation subsets in a 2:1 ratio. The dataset setting accordingly changes to 50% samples as training set, 25% samples as validation set, and 25% samples as testing set. After determine the best hyperparameters, we then retrain the model using the combined training and validation set, and use the remaining 25% as testing set to evaluate the final testing accuracy.

### H.2  THE IMPLEMENTATION SETTINGS FOR COMPARISON STUDIES

To implement our Gradient-based Entire Tree optimization framework, we utilize PyTorch that embeds auto differentiation tools and gradient-based optimizers. Our tree induction method is referred to as `GET` when applied to trees with constant predictions. In cases of trees with linear predictions, we refer to it as `GET-Linear`. It should be noted that `GET` is incorporated our additional subtree polish strategy to mitigate accumulated approximation errors, thereby improving training optimality. In contrast, `GET-Linear` is not equipped with this strategy, as decision trees with linear predictions generally perform well as observed in our experiments. Moreover, decision trees with linear

Table 6: Real-world datasets from UCI and OpenML Repository.

| Dataset Index | Dataset Name | Dataset Size (n) | Feature Number (p) |
|---|---|---|---|
| 1 | airfoil-self-noise | 1,503 | 5 |
| 2 | space-ga | 3,107 | 6 |
| 3 | abalone | 4,177 | 8 |
| 4 | gas-turbine-co-emission-2015 | 7,384 | 9 |
| 5 | gas-turbine-nox-emission-2015 | 7,384 | 9 |
| 6 | puma8NH | 8,192 | 8 |
| 7 | cpu-act | 8,192 | 21 |
| 8 | cpu-small | 8,192 | 12 |
| 9 | kin8nm | 8,192 | 8 |
| 10 | delta-elevators | 9,517 | 6 |
| 11 | combined-cycle-power-plant | 9,568 | 4 |
| 12 | electrical-grid-stability | 10,000 | 12 |
| 13 | condition-based-maintenance_compressor | 11,934 | 16 |
| 14 | condition-based-maintenance_turbine | 11,934 | 16 |
| 15 | ailerons | 13,750 | 40 |
| 16 | elevators | 16,599 | 18 |

predictions are prone to overfitting, a risk potentially exacerbated by subtree polish strategy unless carefully regularized. The tendency of these tree to overfit is also evidenced by results from exiting open-source software for model linear trees, discussed in Appendix H.6. Our methods (GET and GET-Linear) are configured with $N_{epoch} = 3000$ and $N_{start} = 10$, unless otherwise specified.

For benchmarking, the open-source Scikit-learn library in Python is used to implement CART and random forest (RF) methods. The parameter values for these methods are set to default values, unless otherwise specified, such as the specific hyperparameters tuning discussed in Section 5.1. The implementation of HHCART, RandCART and OC1 are adapted from publicly sourced GitHub repository and programmed in Python. We modified their classification-oriented loss functions to adapt for regression tasks. As for the local search method ORT-LS, we reproduce it in Julia due to the absence of open-source code for ORT-LS. The GradTree and SoftDT method are implemented using their respective open-source GitHub repositories, with adjustments made only to the epoch numbers to align with our methods.

Experiments necessitating CPU computation were executed on the high-performance Oracle HPC Cluster, specifically utilizing the "BM.Standard.E4.128" configuration. Each compute node within this cluster is equipped with an "AMD EPYC 7J13 64-Core Processor". Concurrently, experiments requiring GPU resources were conducted on the "Narval" server, which is equipped with an NVIDIA A100 GPU. Additionally, for the comparative analysis of prediction times as elaborated in Section 5.4, we assessed the prediction speeds of each method on the login node of the "Cedar" server.

## H.3 DETAILED TESTING ACCURACY COMPARISON RESULTS ON 16 REAL-WORLD DATASETS

The testing accuracy comparison for specific 16 real-world datasets is detailed in Table 7.

## H.4 RESULTS FOR ALL COMPARED DECISION TREES UNDER FIXED DEPTH SETTING

To figure out the rationale behind the competitive predictive accuracy of our optimized tree, we delve into the analysis of training optimality from the perspective of optimization. As previously discussed, the assessment of the optimization capabilities only make sense when the tree depth is fixed, as the training of tree with different depths corresponds to different optimization tasks. Therefore, the predetermined depths of $D = \{2, 4, 8, 12\}$ are used for training accuracy comparison. Detailed comparisons for all methods in terms of training accuracy, testing accuracy and training time are given in Table 8.

Table 7: The testing accuracy comparison for each dataset.

| Dataset | CART | OC1 | RandCART | HHCART | SoftDT | GradTree | ORT-LS | GET | GET-Linear | RF |
|---|---|---|---|---|---|---|---|---|---|---|
| 1 | 85.27 | 86.38 | 76.79 | 85.71 | 67.06 | 56.46 | 85.24 | 89.21 | 89.96 | 92.58 |
| 2 | 42.14 | 40.53 | 50.70 | 49.05 | 47.46 | 30.47 | 49.51 | 62.07 | 62.07 | 53.22 |
| 3 | 47.29 | 46.15 | 46.94 | 54.75 | 55.28 | 49.68 | 54.20 | 57.29 | 60.24 | 57.64 |
| 4 | 66.50 | 55.39 | 57.24 | 60.61 | 60.80 | 64.07 | 55.62 | 63.38 | 71.59 | 68.10 |
| 5 | 82.19 | 83.20 | 83.30 | 84.43 | 80.81 | 67.03 | 86.39 | 87.97 | 86.79 | 91.00 |
| 6 | 62.36 | 62.71 | 41.64 | 66.95 | 60.31 | 61.53 | 63.68 | 63.68 | 68.12 | 68.38 |
| 7 | 96.98 | 97.23 | 92.06 | 97.15 | 85.36 | 95.30 | 97.59 | 98.01 | 98.23 | 98.27 |
| 8 | 95.89 | 96.25 | 95.78 | 96.25 | 88.66 | 93.69 | 96.65 | 96.93 | 97.06 | 97.63 |
| 9 | 42.56 | 51.53 | 51.83 | 56.32 | 71.60 | 44.22 | 69.57 | 79.82 | 86.67 | 70.48 |
| 10 | 60.19 | 59.03 | 58.42 | 60.91 | 62.76 | 54.76 | 58.52 | 61.66 | 64.26 | 63.16 |
| 11 | 93.33 | 93.15 | 93.11 | 93.55 | 93.16 | 84.66 | 92.84 | 93.86 | 94.04 | 95.92 |
| 12 | 71.17 | 74.72 | 58.08 | 68.80 | 86.35 | 45.01 | 80.66 | 86.76 | 91.19 | 89.59 |
| 13 | 98.58 | 94.37 | 98.45 | 98.63 | 86.83 | 85.17 | 98.93 | 98.98 | 99.98 | 99.50 |
| 14 | 97.34 | 71.14 | 96.11 | 95.23 | 48.37 | 76.81 | 97.78 | 98.11 | 99.97 | 98.74 |
| 15 | 75.96 | 76.55 | 75.48 | 77.92 | 81.61 | 67.49 | 78.44 | 81.18 | 82.53 | 83.24 |
| 16 | 69.12 | 72.30 | 65.03 | 75.66 | 86.57 | 49.73 | 82.61 | 89.41 | 90.88 | 83.59 |

Table 8: Detailed comparison for all compared methods under fixed depth setting.

| Item | D | Greedy Methods | | | | Gradient-based Trees | | State-of-Art Heuristic | Our Tree |
|---|---|---|---|---|---|---|---|---|---|
| | | CART | OC1 | RandCART | HHCART | SoftDT | GradTree | ORT-LS | GET |
| Traing Accuracy (%) | 2 | 47.26 | 49.85 | 33.22 | 46.68 | 49.27 | 39.37 | 66.47 | **71.80** |
| | 4 | 60.90 | 62.90 | 54.62 | 62.59 | 55.81 | 53.34 | 79.51 | **82.40** |
| | 8 | 81.33 | 81.24 | 78.12 | 82.28 | 66.41 | 64.70 | 90.81 | **91.02** |
| | 12 | 93.43 | 92.97 | 93.48 | 94.90 | 72.93 | 66.83 | **97.50** | 96.09 |
| Testing Accuracy (%) | 2 | 46.45 | 48.15 | 32.86 | 46.12 | 48.59 | 38.45 | 64.44 | 70.24 |
| | 4 | 58.60 | 59.76 | 53.14 | 61.24 | 55.27 | 51.40 | 74.84 | 77.89 |
| | 8 | 69.07 | 68.26 | 70.16 | 74.18 | 66.03 | 62.39 | 74.00 | 78.55 |
| | 12 | 67.21 | 64.27 | 63.80 | 67.63 | 72.32 | 63.67 | 67.14 | 71.97 |
| Training Time (s) | 2 | 0.03 | 2648.84 | 0.59 | 3.47 | 1054.35 | 28.74 | 313.89 | 796.65 |
| | 4 | 0.04 | 3439.93 | 1.18 | 5.72 | 1680.10 | 49.24 | 673.39 | 2234.64 |
| | 8 | 0.06 | 3932.06 | 3.46 | 8.87 | 10415.84 | 277.17 | 7872.43 | 2420.84 |
| | 12 | 0.08 | 4179.08 | 9.12 | 15.70 | 173544.28 | 9564.43 | 181308.67 | 9394.67 |

## H.5 ABLATION EXPERIMENTS ON THE STRATEGIES USED IN OUR TREE OPTIMIZATION

The observed improvements in training accuracy of our method GET, as reported in Table 8, can be attributed to two key strategies: the iterative scaled sigmoid approximation and the subtree polish strategy. The comparative results, presented in Table 9, illustrate the impact of these strategies on GET. It should be noted that our method, which utilizes the iterative scaled sigmoid approximation, is referred to as GET in the table. In contrast, when this strategy is not employed, the method is denoted by specific values of scale factors, such as $\alpha = 1$ and $\alpha = 100$. Additionally, the subtree polish strategy is evaluated in the table, with the methods labeled as GET without subtree polish strategy and GET with subtree polish strategy.

Table 9: The effectiveness of strategies used in our tree optimization approach on training accuracy.

| Item | | $D = 2$ | $D = 4$ | $D = 8$ | $D = 12$ |
|---|---|---|---|---|---|
| without Iterative Scaled Sigmoid Approximation (A fixed scale factor) | $\alpha = 1$ (Sigmoid Function) | 55.23 | 65.38 | 81.91 | 93.42 |
| | $\alpha = 100$ | 68.48 | 77.22 | 87.09 | 94.11 |
| with Iterative Scaled Sigmoid Approximation (Our GET method) | GET without subtree polish strategy | 70.86 | 80.30 | 89.45 | 95.16 |
| | GET with subtree polish strategy | 71.80 | 82.40 | 91.02 | 96.09 |

Regarding the iterative scaled sigmoid approximation, we analyze training accuracy with and without this strategy. Without this strategy, we utilized a fixed scale factor for the differentiability approximation, utilizing both the standard sigmoid function with $\alpha = 1$ and a larger scale factor with $\alpha = 100$. The findings indicate substantial improvements with the iterative approach: training accuracy increased by 15.63%, 14.92%, 7.54%, and 1.74% at tree depths of 2, 4, 8, and 12, respectively, when compared to the standard sigmoid approximation. Moreover, when compared to the larger scale factor of $\alpha = 100$, the iterative approach improved training accuracy by 2.38%, 3.08%, 2.36%, and 1.05% for these depths, respectively. These results confirm that our iterative scaled sigmoid approximation surpasses both the standard sigmoid function commonly used in soft decision trees and larger scale factors in differentiability approximation.

The subtree polish strategy also contributes to these improvements, particularly at shallower tree depths. It boosts training accuracy by 0.94%, 2.10%, 1.57%, and 0.93% at tree depths of 2, 4, 8, and 12, respectively.

### H.6 OVERFITTING ISSUE AND COMPARISON FOR TREES WITH LINEAR PREDICTIONS

As discussed in Section 5.5, overfitting issues has been observed with both our method `GET` and `GET-Linear`. Upon further comparison of our `GET-Linear` with the existing open-source library `linear-tree`, it is evident that the overfitting issues are more pronounced in trees with linear predictions. The open-source software `linear-tree` exhibits significantly more severe overfitting issues at depths such as 8 and 12, as detailed in Table 10.

For our method `GET-Linear`, although it benefits from the subtree polish strategy, improving training accuracy by 1% to 2.19%, it conversely results in a reduction of testing accuracy by up to 3.24%. This decrease in testing accuracy for `GET-Linear` begins at relatively small depths, such as 4. Given the susceptibility of trees with linear predictions to overfitting, we opt not to apply the subtree polish strategy to `GET-Linear` in order to mitigate the risk of exacerbating overfitting issues.

Table 10: Comparison of training and testing accuracy for `GET-Linear` and `linear-tree`.

| Depth | Training Accuracy (%) | | | Testing Accuracy (%) | | |
|---|---|---|---|---|---|---|
| | linear-tree | GET-Linear without subtree polish strategy | GET-Linear with subtree polish strategy | linear-tree | GET-Linear without subtree polish strategy | GET-Linear with subtree polish strategy |
| 2 | 79.55 | 81.74 | 82.71 | 79.15 | 79.98 | 80.76 |
| 4 | 84.90 | 84.78 | 86.97 | 72.26 | 81.86 | 80.82 |
| 8 | 93.13 | 88.59 | 89.88 | -400195.15 (overfitting) | 81.50 | 78.26 |
| 12 | 98.28 | 90.88 | 92.74 | -11993680.84 (overfitting) | 64.91 | 49.53 |

To mitigate the overfitting issues for `GET-Linear`, we preliminarily attempt to apply $L_1$ regularization for trainable variables $\boldsymbol{A}$. This approach involves incorporating a regularization term into the loss function $\mathcal{L}$ below, serving to penalize the complexity of the tree structure. The regularized loss is delineated in Equation (5), where $\lambda$ denotes the regularization strength and $\|\cdot\|_1$ represents the $L_1$ norm.

$$\mathcal{L}_{reg} = \sum_{i=1}^{n} \sum_{t \in \mathbb{T}_L} P_{i,t} \left( y_i - (\boldsymbol{k}_t^T \boldsymbol{x}_i + h_t) \right)^2 + \lambda \sum_{t \in \mathbb{T}_b} \|\boldsymbol{a}_t\|_1 \tag{5}$$

Table 11: The comparison for `GET-Linear` with and without regularization across 16 datasets.

| Item | GET-Linear without regularization | GET-Linear with regularization |
|---|---|---|
| Testing Accuracy (%) | 83.24 | 83.97 |

However, identifying the appropriate regularization strength $\lambda$ proves challenging during our experiments, necessitating extensive hyperparameter tuning. This tuning significantly increase the computational cost and the implementation complexity of our method. Consequently, in our experiment reported in Table 1, we did not extensive tune this parameter. We only adjust it between 0 and a very small value $1e-5$ to implement a minimal regularization, aiming to enhance testing accuracy without greatly compromising optimization capabilities. With this slight regularization, the testing accuracy of `GET-Linear` reported in Table 1 is improved by 0.73% compared to the results without regularization, as shown in Table 11. Despite the slight improvement, the overfitting issues for `GET-Linear` still exist, and our preliminary regularization is not sufficient to address this issue. Significant improvements in testing accuracy are achievable through appropriate regularization strategies; however, this requires further exploration and is limited in this paper.

