# OpenReview forum: "Can a Single Tree Outperform an Entire Forest?"
_ICLR.cc/2025/Conference — ICLR 2025 Conference Withdrawn Submission_

### Official Review · Reviewer_tMMM · 2024-11-02

**Soundness:** 2
**Presentation:** 3
**Contribution:** 2
**Rating:** 3
**Confidence:** 5

**Summary:**

This paper proposes a variation of soft (a.k.a.) probabilistic trees that are annealed to obtain "close-to" hard DTs. Additional heuristics, such as subtree polishing is proposed to further enhance the empirical performance. Experiments on small-to-medium scale datasets show competitive accuracy w.r.t. SOTA methods.

**Strengths:**

- Clear and easy to follow algorithm with practical implementation;
- Experiments show convincing results across several baselines (including, surprisingly, random forests). Although the scale of datasets is a concern (see below);

**Weaknesses:**

1. Novelty: I believe the paper combines together several ideas explored in soft DT literature:
- Gradients based learning via sigmoid approximation is well-known approach to train soft trees;
- Based on my understanding, iterative scaled sigmoid approximation is similar to annealing mechanism, which has been previously explored, e.g. in [1] (although this is not the earliest work). Note that [1] also discuss alternative function to sigmoid.
- Subtree polishing reminds weaker version of Tree Alternating Optimization [2,3].
It's good to know that combing these all works quite well, but authors are encouraged to expand on this context and specify differences, if any.

2. Experiments are conducted on small-to-medium datasets where CART is already performing quite well and using RFs are not bringing significant benefits. Authors are encouraged to use more practical high-dimensional (and possibly large-scale) datasets. Moreover, I'd suggest adding [2,3] as additional non-greedy baseline. Note that [3] claim SOTA performance on some datasets used in this paper.


[1] Hussein Hazimeh, Natalia Ponomareva, Petros Mol, Zhenyu Tan, and Rahul Mazumder. ICML (2020). The Tree Ensemble Layer: Differentiability meets Conditional Computation.

[2] M. Carreira-Perpinan and P. Tavallali, 2018. Alternating optimization of decision trees, with application to learning sparse oblique trees.

[3] A. Zharmagambetov and M. Carreira-Perpinan, 2020. "Smaller, More Accurate Regression Forests Using Tree Alternating Optimization".

**Questions:**

- What kind methodology authors use to get the accuracy (%) for regression task? Wasn't able to find that in the paper.
- Author emphasize interpretability as important feature of their methods. Are there any evidences supporting this (other than reporting the structural characteristics, e.g. number of nodes, depth, etc.)?

---

### Official Review · Reviewer_TDNM · 2024-11-03

**Soundness:** 2
**Presentation:** 1
**Contribution:** 1
**Rating:** 3
**Confidence:** 5

**Summary:**

The authors propose a simulated annealing technique to train a soft decision tree, which is ultimately converted into a hard decision tree. The proposed method optimizes soft trees with constant and linear leaves (i.e., linear models within the leaves). Additionally, the method uses several heuristics, such as multiple restarts, randomly selecting simulated annealing steps within a specified range, and optimizing subtrees of all decision nodes of the tree.

**Strengths:**

1. The authors attempt to address the important problem of optimizing hard decision trees, which is an NP-hard problem.
2. The method is evaluated across 16 datasets.

**Weaknesses:**

1. The use of simulated annealing for training soft decision trees lacks novelty (e.g., [1], [2])
2. The "accuracy" metric reported throughout the paper is not formally defined
3. The training time appears exponential in the number of parameters, due to the soft branches routing the entire dataset across all decision nodes ($2^{D + 1} - 1$). This is further exacerbated by the Polish Strategy, which applies the algorithm to all subtrees.
4. It is unclear how a complete binary oblique decision tree can be considered interpretable without incorporating regularization terms for sparsity in decision node weights.

[1] Thomas M. Hehn, Julian F. P. Kooij, and Fred A. Hamprecht. 2020. End-to-End Learning of Decision Trees and Forests. Int. J. Computer Vision 128 (April 2020), 997–1011.

[2] Ajaykrishna Karthikeyan, Naman Jain, Nagarajan Natarajan, and Prateek Jain. 2023. Learning Accurate Decision Trees with Bandit Feedback via Quantized Gradient Descent. Trans. Machine Learning Research (Sept. 2023).

**Questions:**

N/A

---

### Official Review · Reviewer_FRUv · 2024-11-04

**Soundness:** 2
**Presentation:** 3
**Contribution:** 2
**Rating:** 3
**Confidence:** 5

**Summary:**

This paper challenges the common belief that single decision trees cannot match the testing accuracy of random forests, despite the former's advantages in interpretability and lightweight structure. The authors introduce a gradient-based entire tree optimization framework aimed at significantly improving the testing accuracy of oblique regression trees, bringing their performance level close to or even surpassing that of random forests.

**Strengths:**

1.	Your paper is commendably clear and easy to understand.

**Weaknesses:**

1.	The title of the paper is ambiguous. The comparison between trees and random forests requires conditional constraints. Random forest is essentially an ensemble learning framework, in which decision trees are the base learners. Does your title mean that ensemble learning frameworks cannot work on the tree model you proposed? If you are comparing the proposed tree model with the original version of RF, the significance of this comparison is not significant.
2.	In my opinion, this article should focus on the comparison with different oblique decision tree algorithms, especially adding the latest pruning techniques, as this method includes a pruning mechanism.
3.	The method presented in this article lacks a theoretical guarantee. I believe that in a structured model such as a tree model, theoretical explanations would be more convincing than experimental results after parameter tuning.
4.	Tree models still have different structures at the same depth, and Tree depth is not very convincing. It is recommended that the number of nodes in the tree model be displayed.

**Questions:**

See Weakness.

---

### Comment · Area_Chair_wYFM · 2024-11-13
**authors - reviewers discussion open until November 26 at 11:59pm AoE**

Dear authors & reviewers,

The reviews for the paper should be now visible to both authors and reviewers. The discussion is open until November 26 at 11:59pm AoE.

Your AC

---

### Note · Authors · 2024-11-17

I have read and agree with the venue's withdrawal policy on behalf of myself and my co-authors.